# Distribution of Vegetation and Soil Seed Banks Across Habitat Types in Paddy Fields Under Different Farming Practices

**DOI:** 10.3390/plants14020177

**Published:** 2025-01-10

**Authors:** Jeong Hwan Bang, Nan-Hee An, Young-Mi Lee, Jong-Ho Park, Min-Jae Kong, Sung-Jun Hong

**Affiliations:** Organic Agriculture Division, National Institute of Agricultural Sciences, Wanju 55365, Republic of Korea; jhbang0909@korea.kr (J.H.B.); nanhee79@korea.kr (N.-H.A.); youngmi13@korea.kr (Y.-M.L.); jhpark75@korea.kr (J.-H.P.); alswogud@korea.kr (M.-J.K.)

**Keywords:** organic farming, plant distribution, rice paddy, soil seed bank, species richness

## Abstract

Paddy field ecosystems are crucial for crop production, biodiversity conservation, and ecosystem services. Although previous studies have examined paddy field biodiversity, few have addressed how the distribution and species richness of vegetation and soil seed banks are regulated. This study investigated the distribution of wetland plants and soil seed banks in paddy fields across diverse habitat types and identified factors influencing their patterns. Surveys revealed that conventional paddy field habitats contained only a few herbicide-tolerant species (e.g., *Portulaca oleracea* L., *Cardamine flexuosa* With., and *Rorippa palustris* (L.) Besser). In contrast, organic paddy field habitats exhibited higher species richness and abundance. Cluster analysis and nonmetric multidimensional scaling demonstrated that soil seed bank distribution differed markedly in paddy field habitats with different farming practices and was influenced by distinct soil factors. These findings highlight the importance of understanding vegetation and soil seed bank dynamics in paddy field ecosystems to support biodiversity conservation and sustainable agriculture.

## 1. Introduction

Agricultural biodiversity is vital to food systems, providing diverse food sources for humanity [1]. However, intensive agriculture has led to habitat destruction and biodiversity loss, reducing food system resilience [1]. Biodiversity loss has heightened vulnerability to diseases, pests, and climate change, diminishing productivity [2] and threatening human survival [1]. Conserving agricultural biodiversity is essential not only for crop reproduction and productivity [3] but also for mitigating climate change through carbon sequestration and storage [4,5].

Although conventional agriculture has enhanced food production [6], the widespread use of fertilizers and pesticides, along with land-use changes, has caused severe environmental issues. For instance, farmland expansion and synthetic chemical use have degraded habitats, simplified landscapes [7], reduced biodiversity, polluted water and soil, and increased greenhouse gas emissions [8,9,10]. Addressing these issues requires transitioning from conventional land-use practices to sustainable agriculture that conserves biodiversity and enhances ecosystem services [11,12]. According to the Food and Agriculture Organization, organic farming offers a sustainable, ecologically adaptive approach [13,14]. Organic systems emphasize biodiversity conservation, improved energy efficiency, and enhanced water and soil quality while providing various ecosystem services, including climate change mitigation [6,8]. Therefore, further research is required to develop environmentally friendly agroecosystems that address biodiversity loss and climate change.

Soil seed banks are crucial resources for wetland ecosystem regeneration, vegetation restoration, and biodiversity conservation [15,16,17]. A seed bank comprises all viable seeds in or on the soil, characterized by spatial variability [18]. Seeds can remain dormant to survive adverse conditions, aiding population persistence [19]. Plant species differ in their ability to use and retain limiting soil nutrients [20], and plant diversity enhances soil fertility [21]. Previous studies have shown that higher plant diversity strengthens soil resilience, boosts primary productivity, and increases carbon sequestration [22,23,24]. These findings emphasize the critical role of plant diversity in soil nutrient cycling.

Paddy fields constitute 15% of global wetland areas and are vital for biodiversity conservation [25,26]. The present study focuses on a typical Korean rice wetland ecosystem where conventional and organic paddy fields coexist. Despite the growing importance of ecofriendly agriculture in addressing biodiversity challenges, little is known about how farming practices affect soil nutrients and soil seed banks in rice wetlands. Therefore, this study tested two hypotheses by comparing conventional and organic paddy fields: (1) farming practices alter the distribution of paddy field vegetation, and (2) vegetation habitat types markedly influence the physical and chemical properties of paddy soils as well as the distribution of soil seed banks.

## 2. Results

### 2.1. Distribution of Vegetation by Habitat Type

The vegetation survey revealed that plant cover varied by habitat type (Table 1). The footpath area in conventional paddy fields (CP_F) had the highest percentage of bare ground and minimal vegetation. Conversely, the footpath area in organic paddy fields (OP_F) exhibited the highest plant cover, with diverse plant species observed. Both the rice paddy area in conventional paddy fields (CP_R) and the rice paddy area in organic paddy fields (OP_R) contained no other plant species besides *Oryza sativa*.

### 2.2. Distribution of Soil Seed Banks by Habitat Type

Soil seed bank density varied by habitat type (Table 2). Conventional habitats (CP_F and CP_R) showed lower seed bank densities compared with organic habitats (OP_F and OP_R). CP_F had the lowest species count and seed bank density, with sp17 (*Portulaca oleracea*) being the most abundant species (2408 seeds m^−2^). OP_F showed a seed bank density approximately 2.1-fold higher than that of CP_F as well as a more diverse seed bank. In CP_R, sp15 (*Poa annua*) had the highest density (5582 seeds m^−2^). OP_R exhibited the highest soil seed bank density and greater species diversity compared with other habitats, with sp15 (*P. annua*) reaching the highest density of 18,105 seeds m^−2^.

Total seed density and species richness of soil seed banks also differed significantly across habitat types (ANOVA, *p* < 0.05) (Figure 1). Conventional habitats (CP_F and CP_R) exhibited lower total seed densities, whereas organic habitats (OP_F and OP_R) showed higher densities, with OP_R having the highest (Figure 1A). OP_F displayed the highest species richness, whereas CP_F had the lowest (Figure 1B). Species richness did not differ significantly between CP_R and OP_R.

### 2.3. Relationship Between the Distribution of Soil Seed Banks and Soil Environmental Factors

The distribution of soil seed banks in paddy field habitats was assessed using cluster analysis (CA) based on community composition similarity (Figure 2). This analysis identified four distinct clusters (Clusters 1–4). Cluster 1 was primarily composed of OP_R samples, with dominant plant species including sp1 (*Bidens tripartita*), sp7 (*Alopecurus aequalis*), sp12 (*Vicia villosa*), sp13 (*Stellaria alsine*), sp15 (*P. annua*), and sp27 (*Cardamine flexuosa*). Cluster 2 comprised CP_R samples, mainly featuring sp8 (*Erigeron canadensis*) and sp20 (*Eleusine indica*). Cluster 3 contained CP_F samples, with dominant species comprising sp4 (*Senecio vulgaris*), sp14 (*Eragrostis multicaulis*), sp16 (*Rorippa palustris*), sp17 (*P. oleracea*), and sp23 (*Centipeda minima*). Cluster 4 predominantly contained OP_F samples, characterized by diverse species composition, including sp2 (*Galium spurium*), sp3 (*Ranunculus sceleratus*), sp5 (*Oxalis corniculata*), sp6 (*Acalypha australis*), sp9 (*Fimbristylis littoralis*), sp10 (*Digitaria ciliaris*), sp11 (*Lindernia procumbens*), sp19 (*Ludwigia epilobioides*), sp21 (*Cerastium glomeratum*), sp22 (*Mazus pumilus*), sp24 (*Cyperus iria*), sp25 (*Euchiton japonicus*), and sp26 (*Eclipta thermalis*).

Nonmetric multidimensional scaling (NMS) analysis (stress value = 0.009) further reflected differences in soil seed bank composition among habitat types (Figure 3). Paddy field habitats were separated into distinct groups, with significant variations in environmental factors observed among clusters (*p* < 0.05) (Table 3). Cluster 1 (OP_R) and 2 (CP_R) were clustered close together, although Cluster 1 exhibited a higher abundance of soil seed banks (Figure 3A). Conversely, Clusters 3 (CP_F) and 4 (OP_F) were clearly separated. Cluster 3 had few soil seed banks, with the lowest soil water content (WC), organic matter (OM), total nitrogen (TN), and total carbon (TC) values and the highest pH, sodium (Na), iron (Fe), manganese (Mn), and boron (B) levels (Figure 3B). Cluster 4 exhibited the highest WC, OM, TN, TC, and available phosphorus (P_2_O_5_) values, along with the most diverse soil seed bank composition, but the lowest potassium (K), calcium (Ca), and Na levels.

## 3. Discussion

Wetland ecosystems are vital habitats for various organisms and play a key role in biodiversity conservation [27,28]. Vegetation is especially critical for maintaining biodiversity, as it provides a primary habitat [29]. In this study, vegetation surveys revealed distinct wetland habitat characteristics depending on the habitat type (Table 1). CP_F habitats exhibited the highest proportion of bare ground, with minimal vegetation, whereas OP_F habitats showed higher plant cover and greater species diversity. Rice paddy habitats (CP_R and OP_R) were dominated by *O. sativa*, as they are primarily used for crop cultivation. Biodiversity in paddy wetlands can vary based on farming practices [30]. In conventional paddy fields, herbicides are often used for weed control, whereas organic paddy fields rely on biological control, such as golden apple snails (*Pomacea canaliculata*), and mechanical methods, e.g., mowing [31,32]. Consequently, herbicide-sensitive plant species decline in conventional fields, leaving herbicide-tolerant or -resistant species to dominate [33]. In the present study, several species observed in CP_F habitats, such as *D. ciliaris*, *P. oleracea*, *C. flexuosa*, and *R. palustris*, are considered herbicide-tolerant [34]. In contrast, OP_F habitats exhibited greater plant diversity, potentially influenced by grazing snails and/or mowing. These methods are likely less damaging to vegetation compared with chemical control, allowing for a wider variety of species to emerge.

The present study revealed that the species composition and size of soil seed banks varied by habitat type (Table 2). CP_F habitats exhibited patterns similar to the vegetation survey, with the dominance of *E. multicaulis*, *R, palustris*, *P. oleracea*, and *C. minima*. Conversely, OP_F habitats displayed greater diversity and abundance in their soil seed banks. A similar trend was observed between CP_R and OP_R habitats. Soil seed bank richness was highest in OP_F habitats (Figure 1). These results suggest that herbicide use has shaped vegetation type and cover, which, in turn, has affected soil seed bank composition [35,36,37]. The continuous use of herbicides in conventional paddy fields can adversely affect the respiration rate of soil seed banks and seed germination, markedly altering vegetation structure [37,38]. Furthermore, the hydrological factors of paddy wetlands play a crucial role in plant ecological processes. Previous studies have reported that soil water repellency and soil water conservation are important factors for plant establishment and growth [39]. Furthermore, recent studies have also addressed the relationship between vegetation development and environmental factors in terms of ecohydrogeochemical processes [40]; however, these topics are beyond the scope of this study. Vegetation serves as a source for soil seed banks, contributing to their diversity. Soil seed banks act as biodiversity reservoirs, playing a critical role in biodiversity maintenance [41]. With dormancy mechanisms that inhibit seed germination, soil seed banks can endure environmental changes, such as climate change and ecosystem disturbances [42,43]. This resilience makes them essential for biodiversity conservation [42,43,44].

The relationship between soil seed banks and environmental factors across paddy field habitats was clearly defined through CA and NMS analysis (Figure 2 and Figure 3). Significant differences in soil seed bank composition and soil characteristics were observed between habitat types (Table 3). For instance, Clusters 3 and 4 were clearly separated, influenced by distinct soil factors (Figure 3B). Cluster 3, corresponding to CP_F habitats, had a limited number of species (Figure 3A) and minimal vegetation. This habitat exhibited low WC, OM, TN, and TC values but high pH, Na, Fe, Mn, and B levels (Figure 3B). In contrast, Cluster 4, representing OP_F habitats, displayed greater species diversity and abundance, along with high OM, TN, and TC levels. Clusters 1 and 2 were in close proximity, representing the slight overlap between CP_R and OP_R habitats. Notably, soil seed bank distributions differed significantly between conventional (Clusters 2 and 3) and organic (Clusters 1 and 4) habitats (Figure 3A). In conventional habitats, herbicide-resistant plant species were rarely found, whereas organic habitats supported a broader range of species. Previous studies have shown that herbicide use alters the vegetation structure in paddy fields [45], and these changes can affect vegetation succession [31]. Such changes likely harm soil seed banks, reducing plant diversity and genetic variation [33]. Additionally, the present results showed that TC and TN levels were higher in organic habitats (OP_R and OP_F) compared with conventional habitats (CP_R and CP_F). This may be due to the continuous use of fertilizers in conventional paddy fields, leading to nitrogen loss [46]. Conversely, organic farming practices and vegetation presence likely contribute to higher soil carbon levels and nitrogen content [47,48,49]. Overall, these findings confirm that farming practices and habitat types in Korea’s paddy wetlands significantly impact vegetation structure as well as the composition and size of soil seed banks.

The indiscriminate use of pesticides and chemical fertilizers in modern agriculture has increased agricultural productivity but has also caused adverse effects on soil and biodiversity, threatening the sustainability of agriculture [50,51]. Recent studies have reported that pesticide use can lead to soil contamination and pose a threat to biodiversity in cultivated lands [52,53]. For instance, triazine herbicides have significantly impacted benthic algal communities in agricultural ecosystems [54]. Moreover, climate change is expected to increase pest populations [55], raising the likelihood of more frequent pesticide applications and additional pest control measures [56]. The excessive use of synthetic chemicals not only causes severe environmental problems but also poses a threat to human life [57]. Therefore, expanding organic farming practices, such as low-input agriculture, is essential to achieve sustainable farming that allows diverse organisms to coexist [50].

## 4. Materials and Methods

### 4.1. Study Site

The study was conducted on an island located at the southernmost tip of the Korean Peninsula, where paddy fields (34°24′60″ N, 126°17′44″ E) are situated in valleys between low hills and adjacent to the South Sea [34]. The region experiences an annual average temperature of 12 °C–13 °C and receives 1200–1600 mm of precipitation per year, classifying it as a high-rainfall area. The rainy season lasts from June to September, during which approximately 70% of the total annual precipitation occurs (https://jindo.grandculture.net, accessed on 1 December 2024). The study area consists of 40 and 20 ha of conventional and organic paddy fields, respectively, which are geographically clustered. Conventional paddy fields use pesticides and chemical fertilizers, whereas organic paddy fields cultivate rice without synthetic chemicals, relying instead on biological pest control and ecological management practices.

### 4.2. Field Survey

Field surveys were conducted in October 2022 across conventional and organic paddy fields. The study area was divided into two habitat types: the rice and footpath areas [58]. Furthermore, the habitats were categorized according to farming practices into conventional and organic paddy fields. The rice paddy area refers to the rectangular space within footpaths where rice is cultivated under flooded conditions. The footpath area, which borders rice paddies, is constructed with soil to prevent overflow from the rice paddy area. Based on these classifications, the study area was divided into four habitat types (defined in Section 2.1) [34]: CP_R, CP_F, OP_R, and OP_F. Weed management in conventional paddy field habitats involved applying herbicides more than twice annually, and this chemical control method was consistently practiced for over five years. In contrast, organic paddy field habitats utilized golden apple snails (*P. canaliculata*) at a rate of 1.2 kg 10a^−1^ after transplanting rice and physically removed weeds using mowing machines. Additionally, no synthetic chemicals, including herbicides, were used in the organic habitats for seven years. Six random sampling points were selected from each of the four habitat types for vegetation surveys and soil sampling. To conduct vegetation surveys, four 1 × 1 m quadrats were randomly placed at each sampling point, and the vegetation cover was measured. Average cover values were used for analysis. The cover of all plant species, excluding rare species (approximately 1% cover), was recorded to the nearest 5%, and the proportion of bare ground was also documented [27].

### 4.3. Soil Sampling for Soil Seed Bank and Soil Environmental Factor Analyses

Soil samples for soil seed bank analysis were randomly collected at four locations within each quadrat (1 m^2^) using a soil corer (Eijkelkamp BV, Giesbeek, The Netherlands; corer dimensions: 5-cm diameter and 5-cm depth). The four subsamples collected from each sampling point were combined into single composite samples and placed in plastic bags (~400 mL per sample). The composite samples were stored at 4 °C until use in soil seed bank experiments. Additionally, soil samples for analyzing environmental factors (~400 mL per sample) were collected following the same procedure used for soil seed bank sampling.

### 4.4. Evaluation of Soil Seed Bank Characteristics

To evaluate the characteristics of soil seed banks, germination experiments were conducted in a glass greenhouse at the National Institute of Agricultural Sciences (35°49′38″ N, 127°02′30″ E). The greenhouse was equipped with mesh screens to prevent the intrusion of external seeds, and temperature data were recorded hourly using a HOBO Optic USB base station (BASE-U-4, Onset, MA, USA). The soil samples used for the experiment were preprocessed to remove debris, e.g., plant roots, stems, and stones. To create suitable conditions for seed germination, trays (54 × 28 × 5 cm) were filled with commercial growing soil (Pro-100, Chamgrow, Hongsung-gun, Republic of Korea), and the seed bank soil was evenly spread on top in a layer around 1-cm thick [17]. During the 15-week experimental period, the soil was watered at least twice a week to maintain a water level of −2 to 0 cm from the soil surface. The minimum and maximum temperatures recorded in the greenhouse during the experiment were 8.6 °C and 34.6 °C, respectively. Seedlings were identified and their density recorded following the method proposed by [59]. Seed density was calculated as the number of seeds per square meter. Germinated seedlings were identified and removed biweekly, and plants that could not be identified were transplanted into separate pots until identification was possible. All plants that germinated during the soil seed bank experiment were classified and identified using the NATURE database [60].

### 4.5. Soil Analysis

Soil samples were randomly collected from 24 locations (four habitats × six replicates) and preprocessed by air-drying and sieving. The analysis of soil physicochemical properties followed the soil analysis methods outlined by the National Institute of Agricultural Sciences [61]. Soil WC was determined by measuring the weight of the soil before and after drying. Soil pH and electrical conductivity were measured using a pH meter with a glass electrode (Orion 8157BNUMD, Waltham, MA, USA) and a soil:water ratio of 1:5. TN, TC, and OM were analyzed using an elemental analyzer (LECO CN928 Analyzer, St. Joseph, MI, USA) [34]. P_2_O_5_ was measured using the Lancaster method. Exchangeable cations (Exchangeable K^+^, Ca^2+^, Mg^2+^ and Na^+^) were analyzed using an inductively coupled plasma spectrometer (Optima 8300, PerkinElmer, Hopkinton, MA, USA). Ammonium nitrogen was measured using the salicylate method, whereas nitrate nitrogen was determined using an autoanalyzer (QuAAtro, BLTEC Korea, Seoul, Republic of Korea). Micronutrients, such as Fe, Mn, and B, were analyzed using the microwave digestion method.

### 4.6. Statistical Analysis

All statistical analyses were conducted using R software (version 4.2.2) [62], with soil seed banks and environmental factors analyzed. Soil seed bank assemblages from different habitats were classified based on their similarity using cluster analysis (CA) with the unweighted pair group method with arithmetic mean (UPGMA) linkage method and a Bray–Curtis dissimilarity matrix. Nonmetric multidimensional scaling (NMS) was applied to visualize the relationships between sampling points based on Bray–Curtis distances. NMS calculations were performed using the “vegan” package in R [63], and the “metaMDS” function was used to minimize stress values [27]. Additionally, the “envfit” function was employed to assess the relationships between soil seed banks and environmental factors.

## 5. Conclusions

The distribution of vegetation and soil seed banks in paddy wetlands showed significant differences depending on the habitat types influenced by farming practices. The continuous use of herbicides and fertilizers in conventional paddy fields had substantial impacts on the structure of vegetation and potential vegetation (soil seed bank) within the agroecosystem, increasing the spatial heterogeneity of soil physicochemical variables. These effects were particularly evident across habitat types with different flooding conditions. Farming practices in paddy fields can significantly influence the structure and function of paddy ecosystems, serving as key determinants of plant species diversity and genetic diversity. Our findings provide fundamental data that can be utilized for biodiversity conservation and the promotion of sustainable agricultural practices

## Figures and Tables

**Figure 1 plants-14-00177-f001:**
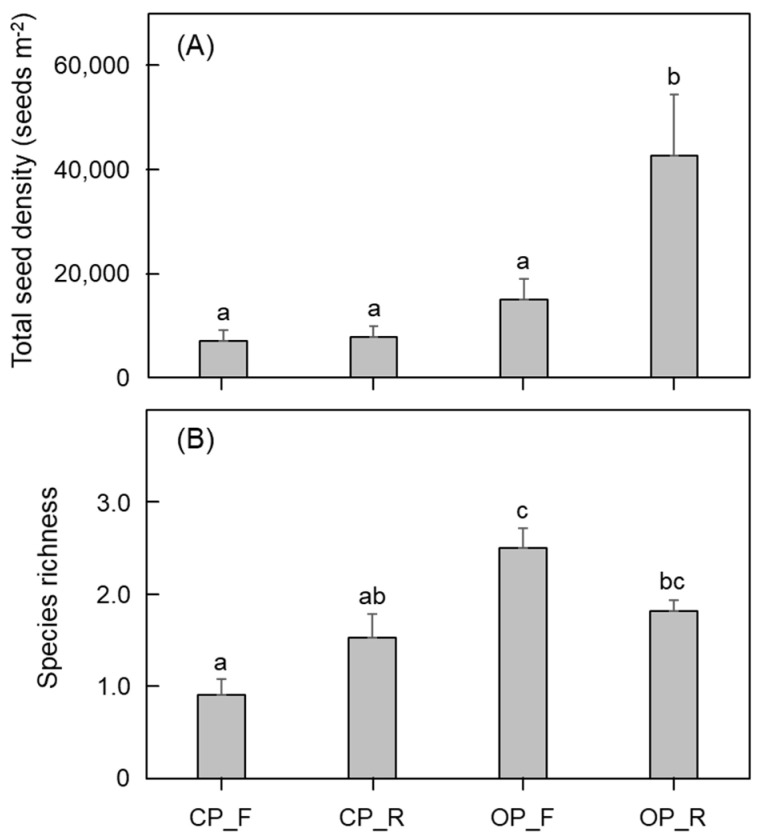
Total seed density (**A**) and species richness (**B**) of soil seed banks by habitat type in paddy fields. CP_F, footpath area in conventional paddy fields; CP_R, rice paddy area in conventional paddy fields; OP_F, footpath area in organic paddy fields; OP_R, rice paddy area in organic paddy fields. The bar graph and error bars represent mean and standard error, respectively. Different letters indicate significant differences between habitat types tested by one-way ANOVA with the Tukey’s post hoc test (*p* < 0.05).

**Figure 2 plants-14-00177-f002:**
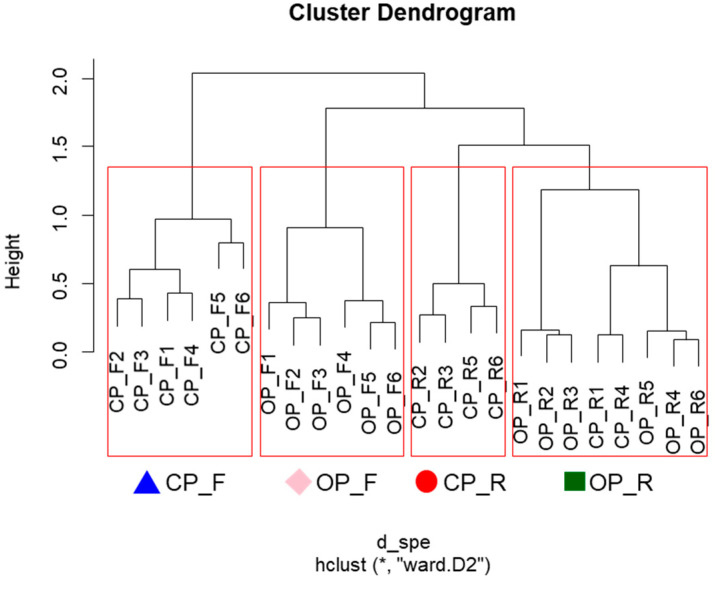
Dendrogram resulting from cluster analysis of soil seed banks in paddy fields. CP_F, footpath area in conventional paddy fields; CP_R, rice paddy area in conventional paddy fields; OP_F, footpath area in organic paddy fields; OP_R, rice paddy area in organic paddy fields.

**Figure 3 plants-14-00177-f003:**
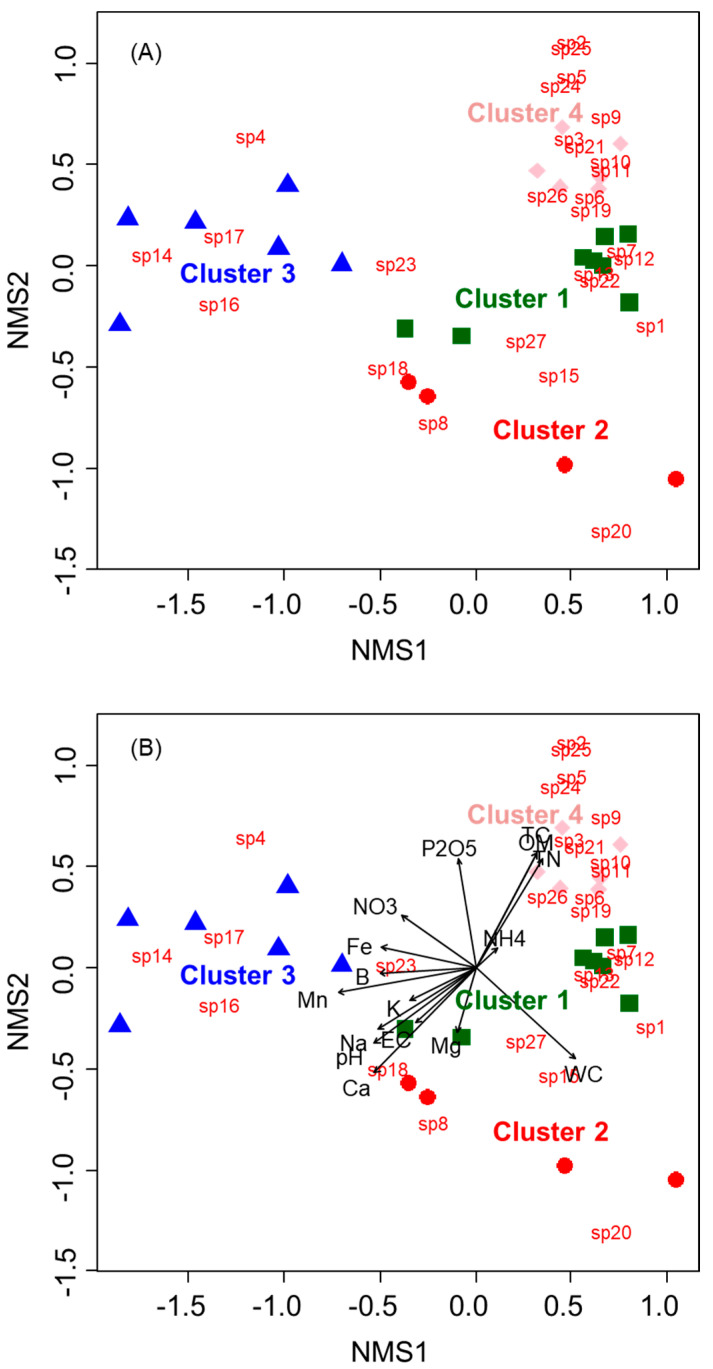
NMS ordination based on Bray–Curtis distances among sampling points of soil seed banks. Ordination with fitted vectors of (**A**) soil seed bank variables and (**B**) soil environmental variables. The shapes of the symbols represent four clusters: squares indicate cluster 1, circles represent cluster 2, triangles correspond to cluster 3, and diamonds signify cluster 4.

**Table 1 plants-14-00177-t001:** Plant cover (%) by habitat type in paddy fields.

	CP_F (%)	CP_R (%)	OP_F (%)	OP_R (%)
Bare ground	97.2 (0.9)	11.7 (1.7)	3.8 (1.2)	19.2 (2.0)
*Oryza sativa* L.	-	88.3 (1.7)	-	80.8 (2.0)
*Digitaria ciliaris* (Retz.) Koeler	0.5 (0.2)	-	72.5 (7.5)	-
*Persicaria lapathifolia* (L.) Delarbre	-	-	9.3 (5.5)	-
*Commelina communis* L.	-	-	6.3 (5.7)	-
*Setaria pumila* (Poir.) Roem. & Schult.	-	-	4.7 (2.2)	-
*Portulaca oleracea* L.	1.7 (0.7)	-	-	-
*Cyperus iria* L.	-	-	1.2 (0.8)	-
*Eclipta thermalis* Bunge	-	-	0.8 (0.2)	-
*Lactuca indica* L.	-	-	0.8 (0.8)	-
*Cardamine flexuosa* With.	0.5 (0.2)	-	-	-
*Echinochloa crus-galli* (L.) P.Beauv. var. *echinatum* (Willd.) Honda	-	-	0.5 (0.2)	-
*Rorippa palustris* (L.) Besser	0.2 (0.2)	-	-	-

Data are presented as Mean (SE). CP_F, footpath area in conventional paddy fields; CP_R, rice paddy area in conventional paddy fields; OP_F, footpath area in organic paddy fields; OP_R, rice paddy area in organic paddy fields.

**Table 2 plants-14-00177-t002:** Soil seed bank density (seeds m^−2^) by habitat type in paddy fields.

No.	Scientific Name	CP_F	CP_R	OP_F	OP_R
sp1	*Bidens tripartita* L.	0 (0)	0 (0)	0 (0)	28 (28)
sp2	*Galium spurium* L. var. *echinospermum* (Wallr.) Desp.	0 (0)	0 (0)	85 (85)	0 (0)
sp3	*Ranunculus sceleratus* L.	0 (0)	0 (0)	28 (28)	0 (0)
sp4	*Senecio vulgaris* L.	57 (57)	0 (0)	0 (0)	0 (0)
sp5	*Oxalis corniculata* L.	0 (0)	0 (0)	170 (139)	0 (0)
sp6	*Acalypha australis* L.	0 (0)	0 (0)	113 (72)	85 (85)
sp7	*Alopecurus aequalis* Sobol.	0 (0)	0 (0)	0 (0)	11,928 (3050)
sp8	*Erigeron canadensis* L.	0 (0)	113 (57)	0 (0)	57 (57)
sp9	*Fimbristylis littoralis* Gaudich.	0 (0)	0 (0)	1587 (684)	57 (36)
sp10	*Digitaria ciliaris* (Retz.) Koeler	0 (0)	0 (0)	2040 (337)	680 (124)
sp11	*Lindernia procumbens* (Krock.) Philcox	0 (0)	0 (0)	708 (217)	368 (52)
sp12	*Vicia villosa* Roth	0 (0)	0 (0)	0 (0)	142 (52)
sp13	*Stellaria alsine* Grimm	0 (0)	170 (62)	1558 (772)	8188 (3829)
sp14	*Eragrostis multicaulis* Steud.	1700 (670)	198 (142)	57 (57)	0 (0)
sp15	*Poa annua* L.	113 (84)	5582 (1858)	907 (327)	18,105 (4465)
sp16	*Rorippa palustris* (L.) Besser	1502 (742)	453 (195)	0 (0)	0 (0)
sp17	*Portulaca oleracea* L.	2408 (762)	227 (84)	85 (38)	0 (0)
sp18	*Cyperus difformis* L.	0 (0)	28 (28)	0 (0)	0 (0)
sp19	*Ludwigia epilobioides* Maxim.	0 (0)	0 (0)	113 (84)	85 (38)
sp20	*Eleusine indica* (L.) Gaertn.	0 (0)	85 (58)	28 (28)	0 (0)
sp21	*Cerastium glomeratum* Thuill.	0 (0)	0 (0)	992 (403)	113 (72)
sp22	*Mazus pumilus* (Burm.f.) Steenis	0 (0)	113 (36)	3938 (1165)	1388 (360)
sp23	*Centipeda minima* (L.) A. Braun & Asch.	1020 (600)	255 (137)	397 (218)	170 (88)
sp24	*Cyperus iria* L.	0 (0)	0 (0)	397 (251)	0 (0)
sp25	*Euchiton japonicus* (Thunb.) Holub	0 (0)	0 (0)	57 (57)	0 (0)
sp26	*Eclipta thermalis* Bunge	113 (57)	85 (58)	1190 (116)	652 (135)
sp27	*Cardamine flexuosa* With.	85 (85)	340 (116)	538 (167)	510 (139)
	Total seed density (seeds m^−2^)	6998	7649	14,988	42,556

Data are presented as Mean (SE). CP_F, footpath area in conventional paddy fields; CP_R, rice paddy area in conventional paddy fields; OP_F, footpath area in organic paddy fields; OP_R, rice paddy area in organic paddy fields.

**Table 3 plants-14-00177-t003:** Relationship between soil factors and NMS ordination of soil seed banks based on analysis via the “envfit” function in R (1000 permutations).

Soil Factors	NMS1	NMS2	r^2^	*p*
Water Content (%)	0.75720	−0.65319	0.6129	0.001
Organic Matter (g kg^−1^)	0.48723	0.87327	0.5493	0.001
pH	−0.81769	−0.57566	0.5498	0.001
EC (dS m^−1^)	−0.74982	−0.66164	0.2234	0.073
TN (%)	0.54882	0.83594	0.5341	0.001
TC (%)	0.48691	0.87345	0.5502	0.001
NH_4_^+^ (mg kg^−1^)	0.75857	0.65159	0.0282	0.736
NO_3_^−^ (mg kg^−1^)	−0.83592	0.54884	0.2828	0.024
P_2_O_5_ (mg kg^−1^)	−0.16212	0.98677	0.3870	0.006
K^+^ (cmol_c_ kg^−1^)	−0.90600	−0.42329	0.1860	0.124
Ca^2+^ (cmol_c_ kg^−1^)	−0.71413	−0.70001	0.7103	0.001
Mg^2+^ (cmol_c_ kg^−1^)	−0.29794	−0.95459	0.1438	0.210
Na^+^ (cmol_c_ kg^−1^)	−0.86108	−0.50847	0.4564	0.005
Fe (%)	−0.97887	0.20450	0.3277	0.016
Mn (mg kg^−1^)	−0.98540	−0.17023	0.6909	0.001
B (mg kg^−1^)	−0.99840	−0.05663	0.3243	0.016

## Data Availability

Data are contained within the article.

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
