# Peer review of "Distribution of Vegetation and Soil Seed Banks Across Habitat Types in Paddy Fields Under Different Farming Practices"

_plants, 2025, doi:10.3390/plants14020177_

Round 1
Reviewer 1 Report
Comments and Suggestions for Authors
Dear Authors,
I am pleased to read this very well written manuscript. The methodology is sound, and the overall presentation of the research is clear. I only have a couple of comments to improve the impact of your work here:
1. Even though I clicked "Yes" for the question "Are the references cited in this manuscript appropriate and relevant to this research?" I think the citation can be improved. For example, some citations are quite old to support your statement, please delete "Dar et al., 2014" (line 155) and add "Wu et al., 2024" and think about to revise some older references.
2. The study site is paddy wetland. Therefore, it is necessary to acknowledge the importance of hydrological factors, show some previous research (e.g., Ruthrof et al., 2019; Van Der Weiden et al. 2023), and state "it is not in the scope of the current work." In this way, readers can have the full picture of the issue and also know that you are aware of this issue.
3. The management recommendation can be enriched in the current discussion. For example, after the documentation of detailed herbicide use, you can extend the topic to a broader view. For example, add Rodriguez et al. (2020) as a general soil conservation to broaden the scope of your work.
I hope my comments are helpful.
References:
1. Rodriguez, A.F., Gerber, S. and Daroub, S.H., 2020. Modeling soil subsidence in a subtropical drained peatland. The case of the everglades agricultural Area. Ecological Modelling, 415, p.108859.
2. Ruthrof, K.X., Hopkins, A.J., Danks, M., O’Hara, G., Bell, R., Henry, D., Standish, R., Tibbett, M., Howieson, J., Burgess, T. and Harper, R., 2019. Rethinking soil water repellency and its management. Plant Ecology, 220(10), pp.977-984.
3. Wu, C.F., Wang, H.H., Chen, S.H. and Trac, L.V.T., 2024. Assessing the efficiency of bird habitat conservation strategies in farmland ecosystems. Ecological Modelling, 492, p.110732.
4. Van Der Weiden, M.J.J., van Haperen, A.M., Kanters, T.J. and Wassen, M.J., 2023. Ecohydrogeochemistry of the Slikken van Flakkee, a former tidal wetland in the Netherlands. Plant Ecology, 224(5), pp.417-434.
Author Response
Responses to Reviewers:
We gratefully thank the reviewers for helpful comments on this manuscript. We hope the relevance and utility of this study is now apparent.
Reviewer #1
Comments 1: Even though I clicked "Yes" for the question "Are the references cited in this manuscript appropriate and relevant to this research?" I think the citation can be improved. For example, some citations are quite old to support your statement, please delete "Dar et al., 2014" (line 155) and add "Wu et al., 2024" and think about to revise some older references.
Response 1: We agree with your comments. The outdated references have been replaced with the following recent research findings in the revised manuscript.
Wu, C.F., Wang, H.H., Chen, S.H. and Trac, L.V.T., 2024. Assessing the efficiency of bird habitat conservation strategies in farmland ecosystems. Ecological Modelling, 492, p.110732.
Bless, A., Davila, F., & Plant, R. (2023). A genealogy of sustainable agriculture narratives: implications for the transformative potential of regenerative agriculture. Agriculture and Human Values, 40(4), 1379-1397.
Duque, T. S., Pinheiro, R. A., Souza, I. M., Silva, G. G., Soares, M. A., & Dos Santos, J. B. (2024). Herbicides and bio-inputs: Compatibility and challenges for sustainable agriculture. Chemosphere, 369, 143878.
Barroso, G. M., Dos Santos, E. A., Pires, F. R., Galon, L., Cabral, C. M., & Dos Santos, J. B. (2023). Phytoremediation: a green and low-cost technology to remediate herbicides in the environment. Chemosphere, 334, 138943.
Osland, M. J., Stevens, P. W., Lamont, M. M., Brusca, R. C., Hart, K. M., Waddle, J. H., ... & Seminoff, J. A. (2021). Tropicalization of temperate ecosystems in North America: The northward range expansion of tropical organisms in response to warming winter temperatures. Global Change Biology, 27(13), 3009-3034.
Yang, Y., Tilman, D., Jin, Z., Smith, P., Barrett, C. B., Zhu, Y. G., ... & Zhuang, M. (2024). Climate change exacerbates the environmental impacts of agriculture. Science, 385(6713), eadn3747.
Comments 2: The study site is paddy wetland. Therefore, it is necessary to acknowledge the importance of hydrological factors, show some previous research (e.g., Ruthrof et al., 2019; Van Der Weiden et al. 2023), and state "it is not in the scope of the current work." In this way, readers can have the full picture of the issue and also know that you are aware of this issue.
Response 2: We agree with your opinion and have added the following sentences to the discussion section. The hydrological factors of paddy wetlands play a crucial role in plant ecological processes. Previous studies have reported that soil water repellency and soil water conservation are important factors for plant establishment and growth (Ruthrof et al., 2019). Furthermore, recent studies have also addressed the relationship between vegetation development and environmental factors in terms of ecohydrogeochemical processes (Van Der Weiden et al., 2023); however, these topics are beyond the scope of this study.
Ruthrof, K. X., Hopkins, A. J., Danks, M., O’Hara, G., Bell, R., Henry, D., ... & Harper, R. (2019). Rethinking soil water repellency and its management. Plant Ecology, 220(10), 977-984.
Van Der Weiden, M. J. J., van Haperen, A. M., Kanters, T. J., & Wassen, M. J. (2023). Ecohydrogeochemistry of the Slikken van Flakkee, a former tidal wetland in the Netherlands. Plant Ecology, 224(5), 417-434.
Comments 3: The management recommendation can be enriched in the current discussion. For example, after the documentation of detailed herbicide use, you can extend the topic to a broader view. For example, add Rodriguez et al. (2020) as a general soil conservation to broaden the scope of your work.
Response 3: We agree with your opinion and have expanded the discussion to a broader perspective by adding a study on herbicide use as follows.
The indiscriminate use of pesticides and chemical fertilizers in modern agriculture has increased agricultural productivity but has also caused adverse effects on soil and biodiversity, threatening the sustainability of agriculture (Bless et al., 2023; Duque et al., 2024). Recent studies have reported that pesticide use can lead to soil contamination and pose a threat to biodiversity in cultivated lands (Tang et al., 2021; Barroso et al., 2023). For instance, triazine herbicides have significantly impacted benthic algal communities in agricultural ecosystems (Lorente et al., 2015). Moreover, climate change is expected to increase pest populations (Osland et al., 2021), raising the likelihood of more frequent pesticide applications and additional pest control measures (Delcour et al., 2015). The excessive use of synthetic chemicals not only causes severe environmental problems but also poses a threat to human life (Yang et al., 2024). Therefore, expanding organic farming practices, such as low-input agriculture, is essential to achieve sustainable farming that allows diverse organisms to coexist (Bless et al., 2023).
Bless, A., Davila, F., & Plant, R. (2023). A genealogy of sustainable agriculture narratives: implications for the transformative potential of regenerative agriculture. Agriculture and Human Values, 40(4), 1379-1397.
Duque, T. S., Pinheiro, R. A., Souza, I. M., Silva, G. G., Soares, M. A., & Dos Santos, J. B. (2024). Herbicides and bio-inputs: Compatibility and challenges for sustainable agriculture. Chemosphere, 369, 143878.
Tang, F. H., Lenzen, M., McBratney, A., & Maggi, F. (2021). Risk of pesticide pollution at the global scale. Nature geoscience, 14(4), 206-210.
Barroso, G. M., Dos Santos, E. A., Pires, F. R., Galon, L., Cabral, C. M., & Dos Santos, J. B. (2023). Phytoremediation: a green and low-cost technology to remediate herbicides in the environment. Chemosphere, 334, 138943.
Lorente, C., Causape, J., Glud, R. N., Hancke, K., Merchan, D., Muniz, S., ... & Navarro, E. (2015). Impacts of agricultural irrigation on nearby freshwater ecosystems: the seasonal influence of triazine herbicides in benthic algal communities. Science of the total environment, 503, 151-158.
Osland, M. J., Stevens, P. W., Lamont, M. M., Brusca, R. C., Hart, K. M., Waddle, J. H., ... & Seminoff, J. A. (2021). Tropicalization of temperate ecosystems in North America: The northward range expansion of tropical organisms in response to warming winter temperatures. Global Change Biology, 27(13), 3009-3034.
Delcour, I., Spanoghe, P., & Uyttendaele, M. (2015). Literature review: Impact of climate change on pesticide use. Food Research International, 68, 7-15.
Yang, Y., Tilman, D., Jin, Z., Smith, P., Barrett, C. B., Zhu, Y. G., ... & Zhuang, M. (2024). Climate change exacerbates the environmental impacts of agriculture. Science, 385(6713), eadn3747.
Bless, A., Davila, F., & Plant, R. (2023). A genealogy of sustainable agriculture narratives: implications for the transformative potential of regenerative agriculture. Agriculture and Human Values, 40(4), 1379-1397.
Thank you for your comment. Revised.

Reviewer 2 Report
Comments and Suggestions for Authors
Review for plants-3400832
The paper in general is an interesting topic, that focuses on the soil seed bank differences in the paddy fields, with different treatment. The methods used in this study are relevant and fine, but what I see as the weakest part is discussion. It should include more literature and authors should compare their findings with the data from literature.
Keywords: must be changed, as too many terms are repeated in the title. Keywords always should be different terms from used in the main title.
Throughout the whole MS, I suggest adding authors of taxa names. Especially in the tables.
Figure 1. I don’t think this graph is very informative, just that there are too many species, and the percentage of some species is too low, to be correctly reflected in the graph.
Line 98-108. If you indicate statistical differences, please write just in the first line where is mentioned and below Fig. 2, what test was used for statistical differences.
Line 109. Paragraph 2.3. I don’t think that authors need to separate each cluster into separate sections like 2.3.1 and etc. Just merge all of them and explain their characteristics in separate paragraphs.
Discussion
Line 168. If these species are herbicide tolerant, please provide a reference to this statement.
Lines 172-188. I don’t understand why authors discuss herbicides and their use in this section and why they pay so much attention to this. From their experiment, herbicide analysis was not included, therefore, I don’t see the connection between their findings and what is discussed here.
Lines 201-204. Again, herbicides are the main issue in this text.
The discussion in general lacks any links to other similar studies, that would compare similar findings. I think this section must be rewritten.
Materials and methods
In these sections, authors should explain each abbreviation (NMS, CA and etc), because it is not clear for the reader what kind of analysis is that. Also, in this section should be included that seed count was expressed by mean values. And I suggest adding standard deviation and include its values into the table 2.
Author Response
Responses to Reviewers:
We gratefully thank the reviewers for helpful comments on this manuscript. We hope the relevance and utility of this study is now apparent.
Reviewer #2
Comments 1: Keywords: must be changed, as too many terms are repeated in the title. Keywords always should be different terms from used in the main title.
Response 1: We agree with your opinion and have revised the Keywords as follows. Key words: organic farming; plant distribution; rice paddy; soil seed bank; species richness
Comments 2: Throughout the whole MS, I suggest adding authors of taxa names. Especially in the tables.
Response 2: We have added the author to the scientific names of plants introduced in the text as follows. Acalypha australis L., Alopecurus aequalis Sobol., Bidens tripartita L., Cardamine flexuosa With., Centipeda minima (L.) A.Braun & Asch., Cerastium glomeratum Thuill., Commelina communis L., Cyperus difformis L., Cyperus iria L., Digitaria ciliaris (Retz.) Koeler, Echinochloa crusgalli (L.) P.Beauv. var. echinatum (Willd.) Honda, Eclipta thermalis Bunge, Eleusine indica (L.) Gaertn., Eragrostis multicaulis Steud., Erigeron canadensis L., Euchiton japonicus (Thunb.) Holub, Fimbristylis littoralis Gaudich., Galium spurium L. var. echinospermum (Wallr.) Desp., Lactuca indica L., Lindernia procumbens (Krock.) Philcox, Ludwigia epilobioides Maxim., Mazus pumilus (Burm.f.) Steenis, Oryza sativa L., Oxalis corniculata L., Persicaria lapathifolia (L.) Delarbre, Poa annua L., Portulaca oleracea L., Ranunculus sceleratus L., Rorippa palustris (L.) Besser, Senecio vulgaris L., Setaria pumila (Poir.) Roem. & Schult., Stellaria alsine Grimm, Vicia villosa Roth
Comments 3: Figure 1. I don’t think this graph is very informative, just that there are too many species, and the percentage of some species is too low, to be correctly reflected in the graph.
Response 3: We agree with your opinion and have removed Figure 1 from the manuscript.
Comments 4: Line 98-108. If you indicate statistical differences, please write just in the first line where is mentioned and below Fig. 2, what test was used for statistical differences.
Response 4: We agree with your opinion. For cases where statistical differences are indicated, we have added the type of test used to determine the statistical differences.
Figure 2. Total seed density (A) and species richness (B) of soil seed banks by habitat type in paddy fields. CP_F, footpath area in conventional paddy fields; CP_R, rice paddy area in conventional paddy fields; OP_F, footpath area in organic paddy fields; OP_R, rice paddy area in organic paddy fields. The bar graph and error bars represent mean and standard error, respectively. Different letters indicate significant differences between habitat types tested by one-way ANOVA with the Tukey’s post-hoc test (P < 0.05).
Comments 5: Line 109. Paragraph 2.3. I don’t think that authors need to separate each cluster into separate sections like 2.3.1 and etc. Just merge all of them and explain their characteristics in separate paragraphs.
Response 5: We agree with your suggestion. Instead of dividing each cluster into separate sections, we have combined them into a single section and described the characteristics of each cluster in separate paragraphs.
The distribution of soil seed banks in paddy field habitats was assessed using cluster analysis (CA) based on community composition similarity (Figure 2). This analysis identified four distinct clusters (Clusters 1–4). Cluster 1 was primarily composed of OP_R samples, with dominant plant species including sp1 (Bidens tripartita), sp7 (Alopecurus aequalis), sp12 (Vicia villosa), sp13 (Stellaria alsine), sp15 (P. annua), and sp27 (Cardamine flexuosa). Cluster 2 comprised CP_R samples, mainly featuring sp8 (Erigeron canadensis) and sp20 (Eleusine indica). Cluster 3 contained CP_F samples, with dominant species comprising sp4 (Senecio vulgaris), sp14 (Eragrostis multicaulis), sp16 (Rorippa palustris), sp17 (P. oleracea), and sp23 (Centipeda minima). Cluster 4 predominantly contained OP_F samples, characterized by diverse species composition, including sp2 (Galium spurium), sp3 (Ranunculus sceleratus), sp5 (Oxalis corniculata), sp6 (Acalypha australis), sp9 (Fimbristylis littoralis), sp10 (Digitaria ciliaris), sp11 (Lindernia procumbens), sp19 (Ludwigia epilobioides), sp21 (Cerastium glomeratum), sp22 (Mazus pumilus), sp24 (Cyperus iria), sp25 (Euchiton japonicus), and sp26 (Eclipta thermalis).
Comments 6: [Discussion] Line 168. If these species are herbicide tolerant, please provide a reference to this statement.
Response 6: We have added references for cases involving herbicide resistance.
In the present study, several species observed in CP_F habitats, such as D. ciliaris, P. oleracea, C. flexuosa, and R. palustris, are considered herbicide-tolerant (Bang et al., 2023).
Bang, J. H., Park, J. H., Lee, Y. M., Chang, C. L., & Hong, S. J. (2023). Comparison of Soil seed bank and Soil characteristics in Conventional Paddy field and Organic Paddy field. Journal of Wetlands Research, 25(4), 237-247.
Comments 7: Lines 172-188. I don’t understand why authors discuss herbicides and their use in this section and why they pay so much attention to this. From their experiment, herbicide analysis was not included, therefore, I don’t see the connection between their findings and what is discussed here. Lines 201-204. Again, herbicides are the main issue in this text.
Response 7: Although our study did not analyse herbicide components, the study sites compared conventional habitats, where pesticides, including herbicides, had been used for an extended period, with organic habitats, where no pesticides were applied. Based on previous research findings and our vegetation survey results, we inferred the impact of herbicide use on paddy vegetation and the soil seed bank as follows:
In this study, the species composition and size of the soil seed bank differed by habitat type (Table 2). The CP_F habitat showed a pattern similar to the vegetation survey results. In contrast, the OP_F habitat exhibited higher diversity and abundance in the soil seed bank. Similar trends were observed in the CP_R and OP_R habitats. These findings suggest that herbicide use influenced vegetation type and coverage, which, in turn, affected the composition of the soil seed bank (Middleton, 2003; Wang et al., 2015; Gomes et al., 2019). From the vegetation survey results of this study (Table 1), the CP_F habitat had the highest proportion of bare ground with almost no vegetation. This indicates that herbicide use in conventional paddy fields affected vegetation. Additionally, the soil seed bank represents potential vegetation, where plant seeds in the soil form vegetation over time. Our soil seed bank study also revealed that plant species appearing in the CP_F habitat were similar to those identified in the soil seed bank, confirming that herbicides significantly impact both vegetation and the soil seed bank.
Middleton, B. A. (2003). Soil seed banks and the potential restoration of forested wetlands after farming. Journal of Applied Ecology, 40(6), 1025-1034.
Wang, G. D., Wang, M., Lu, X. G., & Jiang, M. (2015). Effects of farming on the soil seed banks and wetland restoration potential in Sanjiang Plain, Northeastern China. Ecological Engineering, 77, 265-274.
Gomes, M. P., Richardi, V. S., Bicalho, E. M., da Rocha, D. C., Navarro-Silva, M. A., Soffiatti, P., ... & Sant'Anna-Santos, B. F. (2019). Effects of Ciprofloxacin and Roundup on seed germination and root development of maize. Science of the Total Environment, 651, 2671-2678.
Comments 8: The discussion in general lacks any links to other similar studies, that would compare similar findings. I think this section must be rewritten.
Response 8: We have addressed your comments by including references to similar studies in the discussion section, comparing our findings with those in the literature. The section has been rewritten and revised comprehensively to strengthen its connection to previous research.
Comments 9: [Materials and methods] In these sections, authors should explain each abbreviation (NMS, CA and etc), because it is not clear for the reader what kind of analysis is that. Also, in this section should be included that seed count was expressed by mean values. And I suggest adding standard deviation and include its values into the table 2.
Response 9: Thank you for your valuable feedback. We have provided explanations for each abbreviation (e.g., NMS, CA and etc) in the Materials and Methods section and clarified the analyses used for plant coverage and soil seed bank density.
Thank you for your comment. Revised.

Round 2
Reviewer 1 Report
Comments and Suggestions for Authors
I appreciate the authors' effort to improve the manuscript. I think they have addressed all my concerns and would like to recommend the acceptance of this revised manuscript.
Reviewer 2 Report
Comments and Suggestions for Authors
I thank the authors for the improvements. The manuscript is in much better shape and from my side now can be accepted.